# Far and Near Contrast Sensitivity and Quality of Vision with Six Presbyopia Correcting Intraocular Lenses

**DOI:** 10.3390/jcm11144150

**Published:** 2022-07-17

**Authors:** Miguel Á. Gil, Consuelo Varón, Genis Cardona, José A. Buil

**Affiliations:** 1Ophthalmology Department, Santa Creu and Sant Pau Hospital, Sant Quintí, 89, E08041 Barcelona, Spain; mgila@santpau.cat (M.Á.G.); jbuil@santpau.cat (J.A.B.); 2Department of Optics and Optometry, Universitat Politècnica de Catalunya, Violinista Vellsolà, 37, E08222 Terrassa, Spain; maria.consuelo.varon@upc.edu

**Keywords:** cataract surgery, contrast sensitivity, extended depth of focus, multifocal intraocular lens, quality of vision

## Abstract

The objective of this prospective, randomized, double-masked study was to compare the contrast sensitivity and quality of vision of patients bilaterally implanted with the following six different presbyopia correcting intraocular lenses (IOLs): SV25T0 (*n* = 19), ATLISA 809M (*n* = 18), ATLISA TRI 839MP (*n* = 19), ZKB00 (*n* = 20), ZLB00 (*n* = 20) and Symfony ZXR00 (*n* = 20). For comparison purposes, 36 patients were implanted with a monofocal lens (ZA9003). Contrast sensitivity was assessed binocularly at distance under photopic, mesopic and mesopic plus glare conditions, and at near under photopic conditions. Quality of vision was explored in terms of photic phenomena and spectacle independence. Overall, the monofocal lens offered better contrast sensitivity, under all illumination conditions, and less occurrence and intensity of photic phenomena. Amongst the multifocal IOL (MIOL) designs, the extended depth of focus Symfony ZXR00 provided better contrast sensitivity than the other MIOLs, particularly at intermediate and high spatial frequencies. Up to 40% and 50% of patients implanted with MIOLs reported glare and halos, respectively. The SV25T0 resulted in less occurrence and intensity of halos. The evaluation of photic phenomena and contrast sensitivity under different illumination conditions may reflect real-life, visually challenging situations, and thus provide insightful information to assist ophthalmic surgeons when selecting the best intraocular lens for their patients.

## 1. Introduction

Data from 2015 revealed that 78 percent of US households had a desktop or laptop computer and 75 percent owned at least a handheld device [1]. Given the ubiquity of technology and displays, recent decades have witnessed a progressive shift in the visual needs and demands of the elderly population, with a change in the preference of spectacle independence from near to intermediate distances. Reading text presented on an electronic display is a challenging visual situation in which factors such as size and resolution of visual stimuli, type of task [2] and contrast determine the experience of users. Contrast sensitivity (CS) measurements offer a more complete approach to visual function assessment than that provided solely by high contrast visual acuity (VA). Contrast sensitivity assessment has good sensitivity and specificity for the detection of subtle visual function loss resulting from multifocal intraocular lens (MIOL) implantation [3,4,5,6,7,8,9,10].

Overall, MIOLs have been reported to compromise CS, when compared with monofocal designs [3,4,6]. In addition, performance of MIOLs depends on the lens profile (aspheric vs. spherical), optics (refractive, diffractive or hybrid), add power, and actual light distribution to distance, near and intermediate foci. Amongst MIOLs, diffractive designs have proved superior to refractive MIOLs in terms of CS, and aspheric profiles offer a better performance in challenging situations such as driving at night [8,9,10]. Extended depth of focus (EDOF) designs were introduced to prevent the CS loss encountered with bifocal and trifocal MIOLs [11,12,13,14]. In addition, patients implanted with EDOF tend to report less incidence, size and intensity of halos than those with other multifocal designs [12,13,14,15]. It must be noted that published literature commonly explores CS under photopic and mesopic conditions, which may not necessarily reflect the daily challenges faced by patients. Accordingly, the published recommendations of the American Academy of Ophthalmology Task Force for EDOF MIOLs stress the need to assess CS with and without glare [15].

The aim of the present study was to explore and compare photopic, mesopic and mesopic with glare distance CS, and near photopic CS, as well as quality of vision, of six different presbyopia lenses, including a trifocal and an EDOF design, and a reference monofocal lens, 6 months after lens implantation. A prospective, randomized, double-masked study was designed for this purpose.

## 2. Materials and Methods

### 2.1. Study Sample

Participants were recruited from the Ophthalmology Department of Santa Creu and Sant Pau Hospital, Barcelona, Spain, between February 2019 and March 2020. Inclusion criteria were age over 60 years, bilateral cataract and successful intraocular lens (IOL) implantation, potential VA of 0.1 logMAR or better and preoperative corneal astigmatism equal to 1.25 D or less. Patients with a history of glaucoma, ocular fundus abnormalities, severe dry eye, corneal pathologies and traumatism, irregular astigmatism, corneal or intraocular surgery were excluded. Patients presenting surgical complications (zonular luxation or subluxation, posterior capsular rupture), pupillary trauma, vitreous loss and those cases in which the lens could not be placed in the capsular bag were also excluded from the study. Patients reporting high visual demands, such as frequent nighttime driving, or who were not willing to accept a certain level of post-operative photic phenomena were excluded from the MIOLs groups. In contrast, patients giving preference to excellent vision at distance over the need for spectacle use at near and intermediate distances were included in the monofocal group. Patients manifesting difficulties with examinations and those not attending the follow-up visits were excluded from the study.

All participants provided written informed consent following a full description of the study. The study followed the Declaration of Helsinki tenets of 1975 (as revised in Tokyo in 2004) and received the approval of the Santa Creu and Sant Pau Hospital Ethical Review Board (n. 2211591).

### 2.2. Intraocular Lenses

Six different IOL designs were implanted in this study, and one monofocal lens (Table 1). IOL implantation order was determined with a 1:1:1:1:1:1 block randomization scheme, using the IBM Statistical Package for the Social Sciences (SPSS) software v.27.0 (IBM Corp. Armonk, NY, USA) for Windows. Given a similar sample size for each IOL group, this randomization ratio resulted in an equal allocation of MIOL interventions. Patients were unaware of the type of MIOL they were implanted, although they knew whether their IOLs were monofocal or multifocal. All IOLs (monofocal and multifocal) were provided free of charge to the patients.

### 2.3. Surgical Technique

Surgeries were performed by the same experienced surgeon (M.Á.G.). All surgeries, aimed at bilateral emmetropia and consisted of a 2.75 mm clear corneal incision in the steepest corneal meridian, and a secondary paired incision at 180° if corneal astigmatism was ≥1.00 D. For corneal astigmatisms under 1.00 D, incisions aimed not to introduce cylinder residual errors. Following phacoemulsification, the recommended injectors were employed to place IOLs in the capsular bag. All patients received interventions in both eyes, with a time interval of one week between the interventions.

### 2.4. Contrast Sensitivity

The CSV-1000 contrast sensitivity test (Vector Vision, Inc, Greenville, OH, USA) was employed to assess distance CS binocularly at 2.5 m, under photopic (85 cd/m^2^) (DCSP), mesopic (5 cd/m^2^) (DCSM) and mesopic with glare (DCSMG) conditions. This test consists of a backlit translucent chart presenting four sine-wave grating stimuli corresponding to spatial frequencies of 3, 6, 12 and 18 cycles per degree (cpd) and eight levels of contrast. Measures, in which a four-alternative forced choice paradigm was implemented, were conducted after allowing patients 5 min to adapt to each illumination level. In turn, the Vistech VCTS 6000 system (Vistech Consultants, Inc, Dayton, OH, USA) was used to assess binocular near photopic contrast sensitivity (NCSP) at 40 cm. This test presents five sine-wave grating stimuli sustaining 1.5, 3, 6, 12 and 18 cpd and eight levels of contrast. Ambient illumination was fixed at approximately 120 cd/m^2^, as the Vistech VCTS 6000 is not a backlit test. Patients were permitted small adjustments of their viewing distance, if necessary, to allow for differences in MIOL add power. Near measurements consisted of a two-alternative forced choice paradigm.

Patients used their best distance correction for CS evaluation. For near CS assessment, an additional lens of +2.50 D was used in patients implanted with the monofocal lens, which resulted in partial loss of masking for this IOL group. All measurements were performed by the same experienced, masked optometrist, 6 months after the second intervention.

### 2.5. Quality of Vision

Subjective quality of vision was evaluated by means of a short questionnaire (Appendix A: Quality of vision questionnaire). The aspects under evaluation were spectacle independence for distance, intermediate and near tasks and presence of undesirable photic phenomena such as halos and glare. To ensure a correct and complete interpretation of the questions, patients were shown reference images of halos and glare phenomena.

### 2.6. Data Analysis

The IBM SPSS v.27.0 was used for data analysis. The Kolmogorov–Smirnov test disclosed non-normal distributions of some of the quantitative variables. Therefore, median and range values are reported and, to facilitate comparison, mean and standard deviation (SD) values are also presented. The Kruskal–Wallis test was used for multiple comparisons and, when appropriate, pair-wise comparisons were conducted with the Mann–Whitney test. A *p*-value of 0.05 or less was defined as the cut-off for statistical significance. The DCSP and NCSP values were normalized by dividing the absolute log CS value by the population average reported by Boxer Wachler and Krueger [16] for 3 (2.02 log units), 6 (2.09 log units), 12 (1.85 log units) and 18 (1.45 log units) cpd and photopic conditions.

The estimation of the required sample size was based on previous research on contrast sensitivity with MIOLs in which a threshold for clinical significance was set at a difference larger than 0.15 log units within the same spatial frequency [17]. Considering an α-error of 0.05, a β-error of 0.20 and 7 IOL groups, an initial sample size of 14 participants per group was required to detect 0.15 log unit changes in contrast sensitivity (given a SD of ±0.1 log units).

## 3. Results

### 3.1. Sample Demographics

A total of 152 patients (48 males), age 60 to 86 years, participated in the study. Patients received bilateral and symmetrical implantations of the following IOLs: ATLISA 809M (18 patients), AcrySof ReSTOR SV25T0 (19 patients), Tecnis ZKB00 (20 patients), ATLISA TRI 839MP (19 patients), Tecnis ZLB00 (20 patients), Tecnis Symfony ZXR00 (20 patients) and the monofocal Tecnis ZA9003 (36 patients). Table 2 summarizes demographic data. No statistically significant inter-group differences were found for these variables. All interventions were uneventful and no post-surgical complications were reported. Thus, no patients had to be excluded from the study once the initial allocation was concluded.

### 3.2. Contrast Sensitivity

Photopic, mesopic, mesopic with glare and near photopic CS values for each lens group are summarized in Table 3 (median logarithmic values and range) and shown in Figure 1 (mean logarithmic values). A Kruskal–Wallis analysis revealed statistically significant between-group differences for all spatial frequencies under evaluation and illumination conditions (all *p* ≤ 0.001). Overall, the monofocal ZA9003 offered the best performance at all conditions and spatial frequencies, with statistically significant differences between this lens and all MIOLs, with the exception of the Symfony. Indeed, differences between the ZA9003 and the Symfony reached statistical significance only at certain frequencies (6 cpd DCSP, *p* = 0.003; 12 cpd DCSP, *p* = 0.022; 3 cpd DCSM, *p* = 0.028; 3 cpd DCSMG, *p* = 0.047; 6 cpd DCSMG, *p* = 0.013; 1.5 cpd NCSP, *p* = 0.021; 12 cpd NCSP, *p* = 0.008).

Regarding DCSP, statistically significant pair-wise differences were only found between the Symfony and the other MIOLs, with the Symfony offering better performance at all spatial frequencies, particularly at 12 and 18 cpd. Statistically significant differences were found between the Symfony and the SV25T0 (3 cpd: *p* = 0.002, 6 cpd: *p* = 0.039; 12 cpd: *p* < 0.001, 18 cpd: *p* = 0.005); the ZKB00 (12 cpd: *p* = 0.011); the ZLB00 (6 cpd: *p* = 0.011, 12 cpd: *p* = 0.003, 18 cpd: *p* = 0.008); the ATLISA 809M (6 cpd: *p* = 0.019; 12 cpd: *p* < 0.001, 18 cpd: *p* = 0.002); and the ATLISA TRI 839MP (6 cpd: *p* = 0.009; 12 cpd: *p* = 0.005, 18 cpd: *p* = 0.002).

Similar results were obtained in mesopic conditions, under which the Symfony also proved a superior lens to most of the other MIOLs at intermediate and high spatial frequencies, with differences in the performance of the other MIOLs when compared pair-wise. Statistical differences were found between the Symfony and the SV25T0 (12 cpd: *p* = 0.004, 18 cpd: *p* < 0.001); the ZKB00 (12 cpd: *p* = 0.011, 18 cpd: *p* = 0.014); the ZLB00 (6 cpd: *p* = 0.019, 12 cpd: *p* = 0.001, 18 cpd: *p* = 0.017); the ATLISA 809M (12 cpd: *p* = 0.004, 18 cpd: *p* = 0.003); and the ATLISA TRI 839MP (6 cpd: *p* = 0.030; 12 cpd: *p* < 0.001, 18 cpd: *p* = 0.002).

The Symfony also offered a better performance under mesopic with glare conditions, with statistically significant differences between this lens and the SV25T0 (6 cpd: *p* = 0.008; 12 cpd: *p* < 0.001, 18 cpd: *p* = 0.004); the ZKB00 (6 cpd: *p* = 0.012, 18 cpd: *p* = 0.012); the ZLB00 (6 cpd: *p* = 0.015, 12 cpd: *p* = 0.002, 18 cpd: *p* = 0.028); the ATLISA 809M (6 cpd: *p* = 0.002; 12 cpd: *p* = 0.010, 18 cpd: *p* = 0.006); and the ATLISA TRI 839MP (6 cpd: *p* = 0.007; 12 cpd: *p* = 0.001, 18 cpd: *p* = 0.003).

Finally, for NCSP, the worst performance was obtained with the SV25T0, followed by the ZKB00. Thus, statistically significant differences were found between the SV25T0 and the ZKB00 at 12 cpd (*p* = 0.005), the ZLB00 at 12 cpd (*p* = 0.005) and 18 cpd (*p* = 0.001), the ATLISA 809M at 6 cpd (*p* = 0.032), 12 cpd (*p* < 0.001) and 18 cpd (*p* < 0.001), the ATLISA TRI 839MP at 6 cpd (*p* = 0.047), 12 cpd (*p* = 0.007) and 18 cpd (*p* = 0.005) and the Symfony (*p* < 0.001 at all spatial frequencies except *p* = 0.029 at 3 cpd). In turn, the ZKB00 offered a statistically significant worse performance than the ATLISA 809M at 18 cpd (*p* = 0.020) and the Symfony at 6 cpd (*p* = 0.001) and 18 cpd (0.006). In addition, the Symfony proved to be a superior lens to most of the other MIOLs at NCSP, with statistically significant differences between this lens and the ZLB00 (6 cpd: *p* = 0.015, 12 cpd: *p* = 0.002, 18 cpd: *p* = 0.028); the ATLISA 809M (1.5 cpd: *p* = 0.001; 3 cpd: *p* = 0.013, 6 cpd: *p* = 0.049) and the ATLISA TRI 839MP (1.5 cpd: *p* = 0.004; 6 cpd: *p* = 0.008, 18 cpd: *p* = 0.041). For visualization purposes, Figure 2 shows a comparison of normalized far and near photopic contrast sensitivity values for each lens type.

### 3.3. Quality of Vision

A summary of the results for quality of vision in terms of spectacle independence at far, intermediate and near, halos and glare is shown in Table 4. All parameters under evaluation showed statistically significant differences amongst the groups of lenses. Regarding spectacle independence at far distances, all lenses had a good performance, with the only pair-wise difference arising between the monofocal lens and the SV25T0 (*p* = 0.019), the ZKB00 (*p* = 0.019), the ATLISA 809M (*p* = 0.026) and ATLISA TRI 839MP (*p* = 0.022). At intermediate distances, all MIOLs performed similarly well, and the only statistically significant differences were found between the monofocal lens and the ATLISA 809M (*p* = 0.026) and the ATLISA TRI 839MP (*p* = 0.022). Finally, at near distances the monofocal lens had the worst performance when compared with all MIOLs (all *p* < 0.001). Amongst the MIOLs, the worst performance corresponded to the SV25T0, with pair-wise differences with the ZKB00 (*p* = 0.036), the ATLISA 809M (*p* = 0.005) and the ATLISA TRI 839MP (*p* = 0.004), followed by the ZKB00, with differences between this lens and the ATLISA 809M (*p* = 0.035) and the ATLISA TRI 839MP (*p* = 0.026).

In terms of photic phenomena, the best performance was obtained with the monofocal lens (*p* < 0.05 when compared with all the MIOLs). Amongst the multifocal groups, the best performance was provided by the SV25T0, with statistically significant differences in the occurrence and intensity of halos between this lens and all the other lenses (ZKB00, *p* = 0.013; ZLB00, *p* = 0.020; ATLISA 809M, *p* = 0.003; ATLISA TRI 839MP, *p* = 0.026; Symfony, *p* = 0.009). No statistically significant differences were found between pairs of MIOLs in the presence or intensity of glare.

## 4. Discussion

Patient satisfaction after MIOL implantation is generally good, although quality of vision is often compromised in terms of CS and photic phenomena. In particular, CS may provide better information than other visual function parameters such as high-contrast VA, as a reduction in CS has a negative impact on certain daily tasks, including facial recognition, reading under less than optimal conditions or orientation and mobility in mesopic or scotopic illumination. Paradoxically, however, there is a current lack of consensus regarding instrumentation and methodology to assess CS in patients implanted with MIOLs, as well as on the range of values defining normality [18]. In addition, most studies evaluate only photopic CS [19,20], with scant literature on mesopic [13] and mesopic with glare conditions [21]. Similarly, near CS is seldom explored, and most devices require a specific observation distance, mainly 40 cm, which results in difficulties when comparing MIOLs of different add power. This obstacle was partly resolved in the present study by allowing patients minor adjustments in their observation distance. However, this may lead to a slight overestimation of near CS in MIOLs with high add power such as ZLB00 and ATLISA 809M.

In agreement with published literature, all MIOLs under evaluation resulted in a reduction in CS, when compared with the monofocal group [6,22,23]. This finding has been explained by the distribution of energy to two or more foci required for simultaneous vision [18,24]. Amongst the MIOL groups, the best performance in photopic and mesopic conditions corresponded to the EDOF lens Tecnis Symfony, with results similar to the monofocal lens group for intermediate and high spatial frequencies, in agreement with previous research by Pedrotti and co-workers in photopic conditions [20] and Escandon-García et al. in mesopic conditions [21]. As previously documented, no significant differences were found amongst the other bifocal and trifocal MIOLs in DCSP [6,20] and DCSM [21,25,26]. It must be noted that all explored MIOLs had an aspheric profile, which has been reported to benefit CS in low illumination conditions [27,28]. Regarding mesopic with glare conditions, results were similar to those obtained without glare, with a reduction in CS in all MIOL groups when compared with the monofocal group. However, amongst the MIOLs, the EDOF provided the best results in these conditions, almost comparable with the monofocal lens at intermediate and high spatial frequencies. These findings are partly in disagreement with those reported by previous authors comparing one EDOF design with two trifocal lens designs, in which no differences were encountered between lens groups [21]. Finally, in agreement with published literature, the outcomes for near photopic CS were worse than those obtained in DCSP [4,5], particularly for high spatial frequencies [29]. Amongst the MIOL groups, the best performance corresponded the EDOF Symfony, whereas the SV25T0 and the ZKB00, both low addition lenses, offered the worst results.

It must be noted that all CS measurements were binocular and with patients wearing their best distance correction, to reflect real life conditions. It has been reported that binocular summation may account for a 42% increase in CS [30]. Thus, the present findings may overestimate CS performance, when compared with previous research reporting monocular results. This may partly explain the general lack of differences encountered amongst MIOL groups in terms of CS [31].

Upon exploring spectacle independence at near, as expected, the worst performance corresponded to the monofocal lens [20,32]. Amongst the multifocal designs, the best results were obtained with the ZLB00, ATLISA 809M and ATLISA TRI 839MP. These findings are in disagreement with those reported by Pedrotti and co-workers [20]. In effect, these authors found better results in patients implanted with EDOF and low add power MIOLs (+2.50 D), as compared with a high add power design (+3.00 D).

The evaluation of quality of vision in terms of photic phenomena is very relevant in patients implanted with multifocal lenses. It has been documented that more than 38% of patients reporting unsatisfactory vision mention photic phenomena as the main cause of their difficulties [18]. Previous research is unambiguous in describing a higher incidence of photic phenomena in patients implanted with multifocal designs, when compared with monofocal lenses, with up to 20% patients reporting one or more visual disturbances [23,32]. The present findings give support to the lower incidence of halos and glare in patients implanted with the monofocal lens design. Amongst the multifocal designs, no differences were found in glare occurrence and intensity, with values ranging from 30 to 40% of patients, in agreement with previous research documenting 40% glare in patients implanted with the ATLISA TRI 839MP [33]. The SV25T0 (aspheric, diffractive with refractive periphery, low add power), proved superior to the other MIOLs in the occurrence and intensity of halos. Overall, approximately 50% of patients implanted with multifocal designs reported halos of various intensities, in contrast with published research by Mendicute and co-workers, describing halos in 80% of patients implanted with the ATLISA TRI 839MP [33].

In conclusion, there are many options available to ophthalmic surgeons when selecting the best option for their cataract patients. A careful exploration of the visual requirements and lifestyle of patients is critical to guide lens selection. Monofocal, bifocal, trifocal and EDOF lenses present different advantages, and may offer different quality of vision in challenging conditions. A complete understanding of the best combination of add power, optics and lens design for each particular patient is one the keys leading to patient satisfaction and quality of life.

## Figures and Tables

**Figure 1 jcm-11-04150-f001:**
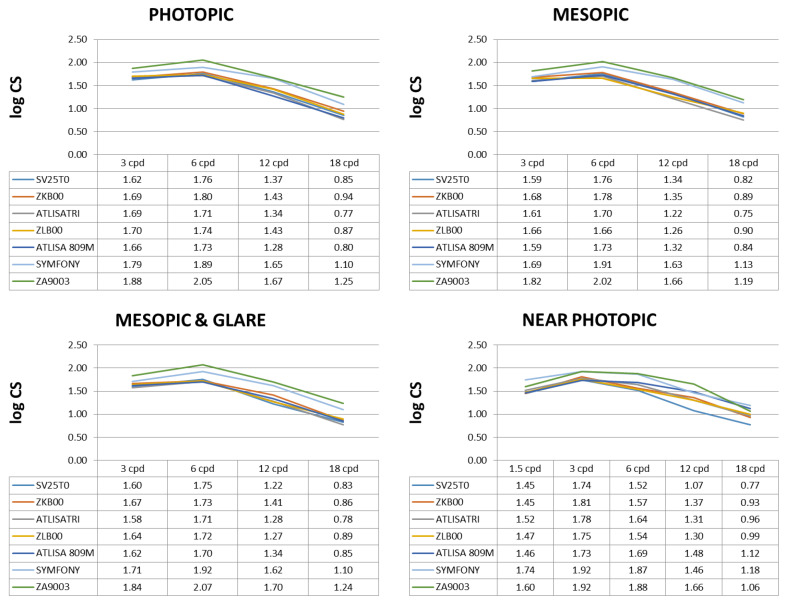
Postoperative binocular corrected distance mean log contrast sensitivity (CS) in photopic, mesopic, mesopic with glare and near photopic conditions.

**Figure 2 jcm-11-04150-f002:**
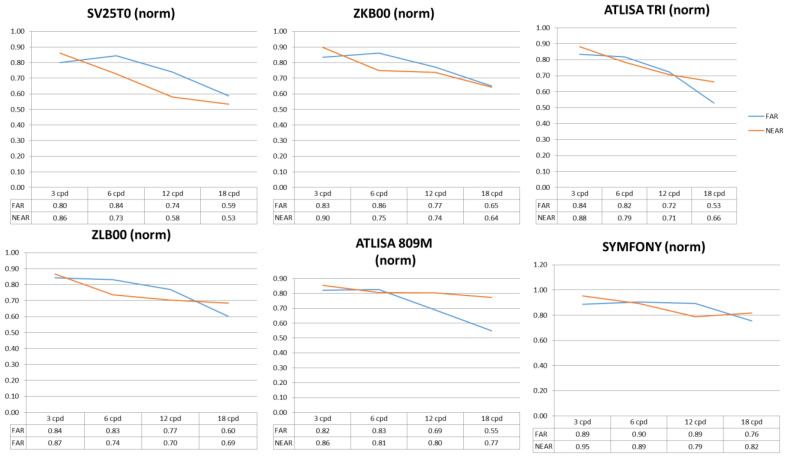
Postoperative binocular corrected photopic distance and near normalized contrast sensitivity (CS) values. The approach reported by Boxer Wachler and Krueger [16] was employed for data normalization.

**Table 1 jcm-11-04150-t001:** Intraocular lenses used in the study, base power of 20.00 diopters (D). Near (n) and intermediate (i) add powers correspond to the plane of the lens. Spherical aberration (SA) is for a 6.0 mm pupil.

LENS	MANUFACTURER	ADD POWER (D)	SA (µm)	OPTICAL DESIGN
AcrySof ReSTOR SV25T0	Alcon Laboratories, Fort Worth, TX, USA	+2.5 (n)	−0.20	Bifocal, anterior aspheric apodized diffractive (3.4 mm) and refractive surface
Tecnis ZKB00	Johnson and Johnson Surgical Vision, Santa Ana, CA	+2.75 (n)	−0.27	Bifocal, anterior aspheric and posterior diffractive surface
Tecnis ZLB00		+3.25 (n)		
ATLISA 809M	Carl Zeiss Meditec AG, Jena, Germany	+3.75 (n)	−0.18	Bifocal, aspheric diffractive
ATLISATri 839MP		+3.33 (n) +1.66 (i)		Trifocal, anterior surface with an aspheric diffractive profile
Tecnis Symfony ZXR00	Johnson and Johnson Surgical Vision, Santa Ana, CA	≈+1.75 (i)	−0.27	Extended depth of focus, wavefront-designed anterior surface, posterior achromatic diffractive surface with echelette design
Tecnis ZA9003	Johnson and Johnson Surgical Vision, Santa Ana, CA	-	−0.27	Monofocal, anterior aspheric

**Table 2 jcm-11-04150-t002:** Demographic data for each lens type. Results are displayed as mean ± standard deviation (SD) or frequency (gender), with the outcome of the ANOVA or the Kruskal–Wallis tests (*p*-value). Pupil diameter was measured under photopic conditions. Lens power and pupil diameter correspond to the right eye.

	SVT250 Bifocal	ZKB00 Bifocal	ZLB00 Bifocal	ATLISA 809M Bifocal	ATLISA Tri 839MP Trifocal	Symfony ZXR00 Extended Depth of Focus	ZA9003 Monofocal	*p*
*n* (eyes)	19	20	20	18	19	20	36	
Age (years)	74.3 ± 7.5	68.9 ± 12.9	73.3 ± 4.6	71.6 ± 7.1	68.7 ± 10.3	68.2 ± 6.2	72.1 ± 5.8	0.064
Gender (male/female)	8/11	5/15	7/13	4/14	4/15	5/15	15/21	0.428
IOL power (D)	21.3 ± 2.4	21.6 ± 3.4	22.3 ± 1.7	22.3 ± 2.4	21.9 ± 4.3	21.8 ± 5.7	21.0 ± 3.6	0.832
Pupil diameter (mm)	3.2 ± 0.6	3.4 ± 0.7	3.2 ± 0.7	3.0 ± 0.6	3.3 ± 0.8	3.3 ± 0.8	3.1 ± 0.7	0.768

**Table 3 jcm-11-04150-t003:** Contrast sensitivity at distance (2.5 m) under photopic (DCSP), mesopic (DCSM) and mesopic with glare (DCSMG), as well as near (33–40 cm) photopic contrast sensitivity (NCSP). Median, maximum and minimum logarithmic values are presented for each lens group and spatial frequency (in cycles per degree, cpd). Also shown are the outcomes of the Kruskal–Wallis test of statistical significance (*p*-value).

	Spatial Frequency	SVT250 Bifocal	ZKB00 Bifocal	ZLB00 Bifocal	ATLISA 809M Bifocal	ATLISA Tri 839MP Trifocal	Symfony ZXR00 EDOF	ZA9003 Monofocal	*p*
DCSP	3 cpd	1.63 1.34–1.93	1.78 1.17–1.93	1.75 1.34–1.93	1.78 1.17–1.93	1.78 1.17–1.93	1.78 1.49–2.08	1.93 1.49–2.08	<0.001
	6 cpd	1.70 1.38–2.29	1.77 1.38–2.29	1.70 1.55–1.99	1.70 1.21–2.14	1.70 1.38–2.14	1.84 1.55–2.29	2.07 1.70–2.29	<0.001
	12 cpd	1.40 0.91–1.69	1.40 0.91–1.99	1.40 1.08–1.69	1.25 0.31–1.84	1.08 0.91–1.84	1.69 1.40–1.99	1.69 0.91–1.99	<0.001
	18 cpd	0.81 0.47–1.25	0.96 0.47–1.55	0.81 0.47–1.25	0.81 0.13–1.10	0.64 0.13–1.25	1.10 0.81–1.55	1.25 0.47–1.55	<0.001
DCSM	3 cpd	1.49 1.34–1.93	1.71 1.17–2.09	1.63 1.34–2.08	1.63 1.17–1.93	1.63 1.34–1.93	1.63 1.34–1.93	1.78 1.63–2.08	0.001
	6 cpd	1.70 1.55–2.29	1.84 1.38–2.14	1.70 0.61–2.14	1.84 1.38–2.14	1.70 1.21–1.99	1.84 1.55–2.29	1.99 1.55–2.29	<0.001
	12 cpd	1.40 0.91–1.69	1.40 0.31–1.69	1.25 0.31–1.69	1.25 0.91–1.84	1.25 0.31–1.69	1.69 1.25–1.99	1.69 0.91–1.99	<0.001
	18 cpd	0.81 0.47–1.10	0.89 0.47–1.25	0.96 0.64–1.25	0.81 0.47–1.40	0.81 0.13–1.25	1.10 0.64–1.55	1.25 0.47–1.55	<0.001
DCSMG	3 cpd	1.56 1.34–1.93	1.63 1.34–1.93	1.63 1.17–2.08	1.63 1.17–1.93	1.49 1.00–1.93	1.78 1.34–1.93	1.78 1.63–2.08	<0.001
	6 cpd	1.70 1.55–2.14	1.70 1.21–2.14	1.84 0.61–2.29	1.70 1.38–1.99	1.70 1.21–2.14	1.99 1.55–2.14	1.99 1.70–2.29	<0.001
	12 cpd	1.40 0.31–1.69	1.40 0.31–1.99	1.25 0.31–1.69	1.40 0.31–1.84	1.25 0.91–1.69	1.54 1.25–1.99	1.69 1.08–1.99	<0.001
	18 cpd	0.81 0.47–1.25	0.89 0.47–1.55	0.96 0.64–1.40	0.81 0.64–1.55	0.64 0.13–1.25	1.10 0.64–1.55	1.25 0.81–1.55	<0.001
NCSP	1.5 cpd	1.54 1.30–1.54	1.54 1.30–1.85	1.54 1.30–1.85	1.54 1.30–1.85	1.54 1.30–1.85	1.85 1.30–2.23	1.54 1.30–2.08	<0.001
	3 cpd	1.64 1.38–1.93	1.93 1.38–1.93	1.64 1.38–2.23	1.64 1.38–2.23	1.64 1.38–2.23	1.93 1.64–2.34	1.93 1.38–2.23	0.001
	6 cpd	1.49 1.32–2.10	1.65 1.04–1.85	1.65 1.04–1.85	1.65 1.32–2.27	1.65 1.32–2.10	1.85 1.32–2.27	1.85 1.32–2.27	<0.001
	12 cpd	1.18 0.90–1.51	1.51 0.70–1.94	1.18 0.70–1.74	1.51 0.90–1.94	1.51 0.90–1.94	1.51 0.90–1.94	1.74 0.90–2.10	<0.001
	18 cpd	0.85 0.60–1.18	0.85 0.60–1.60	1.00 0.60–1.41	1.00 0.60–1.41	1.00 0.60–1.41	1.18 0.60–1.60	1.18 0.30–1.81	<0.001

**Table 4 jcm-11-04150-t004:** Quality of vision for each lens type and results of the Kruskal–Wallis test of statistical significance (*p*-value). All results are percentage of responses.

		SVT250 Bifocal	ZKB00 Bifocal	ZLB00 Bifocal	ATLISA 809M Bifocal	ATLISA Tri 839MP Trifocal	Symfony ZXR00 EDOF	ZA9003 Monofocal	*p*
Spectacle use at far	Always	0	0	0	0	0	5.6	11.1	0.002
	Sometimes	0	0	5.0	0	0	5.6	13.9	
	Never	100.0	100.0	95.0	100.0	100.0	88.9	75.0	
Spectacle use at intermediate	Always	0	0	0	0	0	5.6	11.1	0.033
	Sometimes	10.5	5.3	10.0	0	0	0	13.9	
	Never	89.5	94.7	90.0	100.0	100.0	94.4	75.0	
Spectacle use at near	Always	15.8	5.3	5.0	0	0	16.7	75	<0.001
	Sometimes	52.6	52.6	30.0	23.5	22.2	22.2	25.0	
	Never	31.6	42.1	65.0	76.5	77.8	61.1	0	
Halos occurrence and intensity	None	84.2	42.1	50.0	35.3	50.0	38.9	94.4	<0.001
	1	5.3	26.3	15.0	11.8	5.6	16.7	5.6	
	2	5.3	21.1	0	17.6	27.8	44.4	0	
	3	5.3	10.5	35.0	35.3	16.7	0	0	
Glare occurrence and intensity	None	0	0	0	0	5.6	0	0	0.016
	1	47.4	52.6	47.4	35.3	16.7	38.9	77.8	
	2	15.8	5.3	31.6	17.6	27.8	22.2	16.7	
	3	21.1	21.1	5.3	29.4	11.1	5.6	0	

## Data Availability

The datasets generated and analyzed during the study are available from the corresponding authors upon reasonable request.

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
