# Peer review of "Far and Near Contrast Sensitivity and Quality of Vision with Six Presbyopia Correcting Intraocular Lenses"

_jcm, 2022, doi:10.3390/jcm11144150_

Round 1

Reviewer 1 Report

1. What additional value the present study has imparted to existing knowledge? Most, nearly all, things about various lens design have already published. 

2. Randomization process need elaboration in method section? How blind allocation was ensured? Were patients aware about type of lens being implanted? How ethical issue associated with choice of choosing IOL was tackled?

3. Demographics should also provide information about visual requirements of participants, as this might affect their response in questionnaire.  

4. Questionnaire used to evaluate the quality of vision should be part of manuscript provided as appendix to readers.

Author Response

  1. What additional value the present study has imparted to existing knowledge? Most, nearly all, things about various lens design have already published. 

ANSWER: Although the performance of the diverse IOL designs explored in the present study has already been investigated by other authors, published literature commonly describes and compares the performance of two or three of these IOLs, with or without the inclusion of a monofocal group. In addition, most studies are not double masked and explore contrast sensitivity solely under photopic conditions. In contrast, the present research was double masked and contrast sensitivity was also investigated under mesopic, mesopic with glare and near distance conditions, which may reflect many of the daily challenges encountered by these patients. We believe that these additions to the body of literature on MIOLs may be relevant to practitioners to assist them in the best lens selection for their patients.

  1. Randomization process need elaboration in method section? How blind allocation was ensured? Were patients aware about type of lens being implanted? How ethical issue associated with choice of choosing IOL was tackled?

ANSWER: These aspects are fully described in the revised manuscript. Briefly, even though the surgeon who conducted all interventions was aware of the type of lens being implanted, all post-operative contrast sensitivity and quality of vision measurements were conducted by the same optometrist, who was masked to the type of lens implanted to each patient. However, masking was not possible for patients implanted with a monofocal IOL, as the addition of an external +2.50 D lens was required for contrast sensitivity measurements. In addition, patients were not aware of the type of multifocal lens they had been implanted with, with the exception, once again, of patients implanted with a monofocal lens.

The reviewer is correct in noting the importance of ethical considerations. For intraocular lens implantation, it is critical to understand the visual requirements of patients, in order to select the best possible lens design to meet these requirements (in terms of multifocality, add power, asphericity, etc.). As the purpose of the study was precisely to explore the performance of these relatively new lens designs, it was not possible to provide patients with prior information on lens performance. However, all patients implanted with multifocal lenses manifested their preference for spectacle independence at the various distances and, at the same time, did not report demanding visual requirements at any of these distances, as it is well known that multifocal options offer a compromise between spectacle independence and visual quality. All patients were informed of these aspects prior to the intervention and all of them signed an informed consent. Patients implanted with monofocal lenses were those manifesting their preference for excellent vision at distance, even at the cost of requiring spectacles at near or intermediate distances. All these considerations received the approval of the corresponding ethical committee.   

Regarding randomization, given a similar sample size for each MIOL group, a 1:1:1:1:1:1 scheme results in an equal allocation of MIOL interventions.

  1. Demographics should also provide information about visual requirements of participants, as this might affect their response in questionnaire. 

ANSWER: Unfortunately, as noted above, visual requirements were not documented beyond the distinction between those patients giving more importance to spectacle independence than to quality of vision (implanted with multifocal lenses) and those preferring excellent visual quality at distance even if requiring spectacles for near and intermediate tasks (implanted with monofocal lenses). 

  1. Questionnaire used to evaluate the quality of vision should be part of manuscript provided as appendix to readers.

ANSWER: An English translation of the questionnaire has been added to the manuscript as an appendix. Thank you for this suggestion.

Reviewer 2 Report

This paper compared contrast sensitivity and quality of vision with different lens designs. Although all of the lenses have been investigated before, this paper directly compared these designs with each other under various lighting and glare conditions.

It is a well written paper, that is easy to understand and generally clear in meaning (there are one or two exceptions where further clarity is required eg lines 82-87).  

It would be useful to expand on the methodology a little in places such as the process of randomisation, and also more detail on the questionnaire used.

The conclusions are consistent with the evidence and address the main question. 

Line 82-87: please could you re-phrase this, in order to improve the flow of this section.

Line 103: "..consisted of a"

Author Response

  1. This paper compared contrast sensitivity and quality of vision with different lens designs. Although all of the lenses have been investigated before, this paper directly compared these designs with each other under various lighting and glare conditions.

ANSWER: Thank you. We believe that these aspects of our study were novel and provide sufficient evidence to warrant publication.

  1. It is a well written paper, that is easy to understand and generally clear in meaning (there are one or two exceptions where further clarity is required e.g., lines 82-87).   

ANSWER: We have rewritten and added some information to these sentences for clarity.

  1. It would be useful to expand on the methodology a little in places such as the process of randomisation, and also more detail on the questionnaire used. 

ANSWER: We have added relevant information to the revised methodology section of the manuscript and provided the quality of vision questionnaire as a supplementary file. Regarding randomization, given a similar sample size for each MIOL group, a 1:1:1:1:1:1 scheme results in an equal allocation of MIOL interventions.

  1. The conclusions are consistent with the evidence and address the main question. 

ANSWER: Thank you for this kind appraisal.

  1. Line 82-87: please could you re-phrase this, in order to improve the flow of this section.

ANSWER: We have rewritten this section of the manuscript: “Six different IOL designs were implanted in this study, and a monofocal lens (Table 1). IOL implantation order was determined with a 1:1:1:1:1:1 block randomization scheme, IBM Statistical Package for the Social Sciences (SPSS) software v.27.0 (IBM Corp. NY, US) for Windows. Given a similar sample size for each IOL group, this randomization ratio results in an equal allocation of MIOL interventions. Patients were unaware of the type of MIOL they were implanted, although they knew whether their IOLs were monofocal or multifocal. All IOLs (monofocal and multifocal) were provided free of charge to the patients.”

  1. Line 103: "..consisted of a"

ANSWER: We have edited this. Thank you for pointing this grammatical error to us.

Reviewer 3 Report

Thanks the authors for conducting this randomised controlled trial on presbyopia intraocular lens

Firstly, please spell out all abbreviations at their first time appearance in text. E.g. in Abstract, MIOL was not spelt out in full. Does it mean myopic intraocular lens, or monofocal intraocular lens, or multifocal intraocular lens?

And SPSS was spelt out in full only in line 115, but not in line 80. They should be reversed.

Please mention the sample size calculation methods for this block randomisation in 1:1:1:1:1:1 of patents. There should be one sample size calculation for contrast sensitivity study, and another one for questionnaire based part of the study.
Please mention the period of this study. When was recruitment done?

For patients with only one eye having cataract, were they included or excluded in this study? What if after aiming for emmetropia for one eye, the other non cataract eye became anisometropic?

Were all patients having binocular cataract surgery?  If so, were they done on the same day, or sequential cataract surgery separate by days or weeks? Were both eyes aimed at emmetropia or with mini mono vision, or -0.5D difference?

Please mention how many patients were excluded from this study according to the different exclusion criteria.

Line 127 is a bit confusing, as the statement quoting the number of subjects participated in this study was only counting the intervention groups, but excluding the control group. It would be better to include also the control group subjects so the total number would truly reflect the total subjects analysed.

Please mention the financial disclosure since this study included commercial products from different companies. Did patients need to pay for these premium intraocular lens? And the price difference across different implanted intraocular lens.

Author Response

  1. Thanks the authors for conducting this randomised controlled trial on presbyopia intraocular lens.

ANSWER: We would like to express our gratitude to the reviewer for providing these comments and suggestions.

  1. Firstly, please spell out all abbreviations at their first time appearance in text. E.g. in Abstract, MIOL was not spelt out in full. Does it mean myopic intraocular lens, or monofocal intraocular lens, or multifocal intraocular lens? And SPSS was spelt out in full only in line 115, but not in line 80. They should be reversed.

ANSWER: Thank you for noting these errors, which we have corrected in the revised manuscript.

  1. Please mention the sample size calculation methods for this block randomisation in 1:1:1:1:1:1 of patients. There should be one sample size calculation for contrast sensitivity study, and another one for questionnaire-based part of the study.

ANSWER: Given that the questionnaire included only nominal data, sample size calculations were conducted only for the contrast sensitivity part of the study. We have added this information to the revised Data Analysis section of the manuscript: “The estimation of the required sample size was based on previous research on contrast sensitivity with MIOLs in which a threshold for clinical significance was set at a difference larger than 0.15 log units within the same spatial frequency (Ginsburg AP. Contrast sensitivity and functional vision. Int Ophthalmol Clin 2003;43:5–16). Considering an α-error of 0.05, a β-error of 0.20 and 7 IOL groups, an initial sample size of 14 participants per group was required to detect 0.15 log unit changes in contrast sensitivity (given a SD of ±0.1 log units).”

  1. Please mention the period of this study. When was recruitment done?

ANSWER: All patients were recruited between February 2019 and March 2020. We have added this information in the manuscript.

  1. For patients with only one eye having cataract, were they included or excluded in this study? What if after aiming for emmetropia for one eye, the other non-cataract eye became anisometropic?

ANSWER: Only patients with successful bilateral implantation of either monofocal lenses or MIOLs were included in the study, and all interventions aimed at emmetropia. This information is highlighted in the revised manuscript.

  1. Were all patients having binocular cataract surgery?  If so, were they done on the same day, or sequential cataract surgery separate by days or weeks? Were both eyes aimed at emmetropia or with mini mono vision, or -0.5D difference?

ANSWER: As noted above, all interventions were bilateral and aimed at emmetropia in both eyes. All procedures were conducted with a separation of one week between interventions, and measurements were conducted six months after the second intervention. We have added this relevant information to the Methods Section of the manuscript.

  1. Please mention how many patients were excluded from this study according to the different exclusion criteria.

ANSWER: The strict inclusion criteria, and the fact that all interventions were uneventful, with successful implantation of the lens in the capsular bag and without any post-surgical complication, resulted in no patients being excluded from the study, once allocation to the diverse MIOL groups was conducted. We have added the information regarding uneventful interventions to the manuscript.

  1. Line 127 is a bit confusing, as the statement quoting the number of subjects participated in this study was only counting the intervention groups, but excluding the control group. It would be better to include also the control group subjects so the total number would truly reflect the total subjects analysed.

ANSWER: We agree with the reviewer. We have edited this paragraph for clarity of presentation: “A total of 152 patients (48 males, age 60 to 86 years) participated in the study. Patients received bilateral and symmetrical implantations of the following IOLs: ATLISA 809M (18 patients), AcrySof ReSTOR SV25T0 (19 patients), Tecnis ZKB00 (20 patients), ATLISA TRI 839MP (19 patients), Tecnis ZLB00 (20 patients), Tecnis Symfony ZXR00 (20 patients) and the monofocal Tecnis ZA9003 (36 patients).”

  1. Please mention the financial disclosure since this study included commercial products from different companies. Did patients need to pay for these premium intraocular lens? And the price difference across different implanted intraocular lens.

ANSWER: We have added the following information to the financial disclosure: “None of the authors had any commercial interest with any of the products included in the study. None of the authors received any funding from companies related to the products included in the study”.

All IOLs were provided free of charge to the patients, both monofocal and premium MIOLs. Patients were unaware of the actual cost of each lens, should they had been asked to pay for the intervention. We have also added this information to the revised manuscript.

Round 2

Reviewer 1 Report

The manuscript may be accepted in present for consideration for publication.